# The Role of Side Chains in the Fine-Tuning of the Metal-Binding Ability of Multihistidine Peptides

**DOI:** 10.3390/molecules27113435

**Published:** 2022-05-26

**Authors:** Enikő Székely, Gizella Csire, Bettina Diána Balogh, Judit Zsuzsa Erdei, Judit Mária Király, Judit Kocsi, Júlia Pinkóczy, Katalin Várnagy

**Affiliations:** Department of Inorganic and Analytical Chemistry, University of Debrecen, H-4032 Debrecen, Hungary; szekely.eniko@science.unideb.hu (E.S.); csiregizus@gmail.com (G.C.); balogh.bettina@science.unideb.hu (B.D.B.); jutka.erdei@med.unideb.hu (J.Z.E.); kiralyj91@gmail.com (J.M.K.); juditkocsi88@gmail.com (J.K.); julz9428@gmail.com (J.P.)

**Keywords:** multihistidine peptides, copper(II), nickel(II), zinc(II), complex, stability constant, electrochemical parameters

## Abstract

The systematic studies of copper(II), nickel(II) and zinc(II) ion complexes of protected multihistidine peptides containing amino acids with different side chains (Ac-SarHAH-NH_2_, Ac-HADH-NH_2_, Ac-HDAH-NH_2_, Ac-HXHYH-NH_2_ X, Y = A, F, D or K, Ac-HXHAHXH-NH_2_, X = F or D) have provided information about the metal ion and protein interaction and have made it possible to draw conclusions regarding general trends in the coordination of metal complexes of multihistidine peptides. The stability of the metal complexes significantly depends on the position of the histidines and amino acids, which are present in the neighbourhood of the histidine amino acids as well. The most significant effect was observed on peptides containing aspartic acid or phenylalanine. The redox parameters of complexes, however, depend on the number and position of histidines, and the other side chain donor atoms have practically no effect on the electrochemical properties of imidazole-coordinated species. However, the presence of aspartic acid side chains results in a more distorted geometry of amide-coordinated species and increases the reducibility of these complexes.

## 1. Introduction

The CuZnSOD enzyme isolated from bovine erythrocytes has been most intensively studied, and both the complete amino acid sequence [1] and the X-ray structure are known [2]. The structural analysis of the enzyme has proven that the copper(II) ion is coordinated through four histidine imidazole nitrogens, while the zinc(II) ion is bound through three histidine imidazole nitrogens and the carboxylate group of the aspartic acid. These two binding sites are connected by an imidazolato bridge [3,4].

Our research work focuses on mimicking the active site of the CuZnSOD enzyme. In order to model the active site of this enzyme, terminally protected peptides containing varying numbers of histidine in different positions were synthesized and studied. The general trends concerning the fine-tuning effect of the change in distance between histidine residues were studied for the tetra-, penta-, hexa- and heptapeptides containing 2–4 histidine residues, and these results were summarized in some previous papers [5,6,7,8,9,10,11,12,13] and reviews [14,15].

The N- and C-terminally protected multihistidine peptides may mimic the copper(II)-containing active site of the CuZnSOD enzyme, since the imidazole-coordinated [CuL] and [CuL_2_] complexes are structurally similar to the binding site. These complexes are formed at slightly acidic pH, but an increase in pH results in the deprotonation and coordination of peptide amide donor groups of the peptide backbone and [Cu_x_L_y_], [Cu_x_H_−1_L_y_], [Cu_x_H_−2_L_y_], [Cu_x_H_−3_L_y_] (x = 1, 2, (3), y = 1, 2) exist in physiological and slightly basic solutions depending on the number of histidines in the ligand.

Numerous multihistidine peptides in which the histidines are separated by one, two or more amino acid were investigated, and the results were published in the aforementioned papers [5,6,7,8,9,10,11,12,13,14,15,16].

For peptides with the Ac-HXH-NH_2_ sequence, the two histidines are separated by one amino acid. Two and four imidazole-coordinated [CuL] and [CuL_2_] complexes are formed. In parallel with the increase in pH, the cooperative deprotonation and coordination of two amide nitrogens take place, and a [CuH_−2_L] species is formed, in which the [N_Im_,N^−^,N^−^,N_Im_] donor set binds the copper(II) ion, leading to the formation of (7,5,6) membered joined chelates [5].

The number of amino acids between histidines increases in the Ac-HXXH-NH_2_ ligands. The imidazole nitrogen of the C-terminal histidines behaves as an anchor group. Following the formation of stable imidazole-coordinated [CuL] and [CuL_2_] complexes the deprotonation of amide nitrogens takes place in separate steps, and [CuH_−1_L], [CuH_−2_L], [CuH_−3_L] are present in the alkali pH range. In these species, the ligand is bound to the copper(II) ion through [(N^−^)_x_,N_Im_] (x = 1, 2 or 3) donor groups, and the imidazole group of the N-terminal histidines may contribute to metal binding in the [CuH_−1_L], [CuH_−2_L] complexes [12,13].

In the Ac-HXHZH-NH_2_ peptides, one more histidine is built in the peptide chain. In slightly acidic mediums, the [CuL] complex is the major species with the formation of a macrochelate via three imidazole rings. The existence of these macrochelate rings suppresses but does not prevent the deprotonation of amide nitrogens. In the physiological pH range, the [CuH_−2_L] complex predominates, and the coordination mode is [N_Im_,N^−^,N^−^,N_Im_]. At a higher pH range, the [CuH_−3_L] complex is formed with coordination of three amide nitrogens and one imidazole nitrogen, resulting in [N^−^,N^−^,N^−^,N_Im_] coordination mode [8].

For the Ac-HXXHZH-NH_2_ and Ac-HGGGHGH-NH_2_ peptides, which also contain three histidines, the ligand coordinates the metal ion in the [CuL] complex via three imidazole nitrogens, and similarly to other multhistidine peptides, the [CuH_−2_L] and [CuH_−3_L] complexes are formed in parallel with the increase in pH [13].

In the case of all the aforementioned three-histidine-containing peptides, both the C-terminal and intermediate histidines can behave as anchor groups in the metal ion-induced deprotonation and coordination of peptide nitrogens that results in the formation of coordination isomers of [CuH_−2_L] and [CuH_−3_L] species. It was generally concluded that although the ratio of coordination isomers depends on the amino acid sequence of the peptides, the binding of metal ions in the [CuH_−2_L] complex of all peptides as well as in the [CuH_−3_L] complexes of the Ac-HXHZH-NH_2_ peptides is more preferred at the C-terminus, while in the [CuH_−3_L] complex of hexa- and hepta-peptides, the intermediate histidine is the more favourable binding site [8,13].

Due to the presence of more histidines in the molecules, polynuclear complexes are also formed in solutions containing an excess of copper(II) ions. The predominant dinuclear complex is [Cu_2_H_−4_L] in the pH range of 7–9, in which the C-terminal histidine binds one metal ion and the intermediate histidine the other with the [(N_Im_),N^−^,N^−^,N_Im_] coordination mode. At higher pH, the dinuclear [Cu_2_H_−5_L] complex was also detected with both [N^−^,N^−^,N^−^,N_Im_] and [N^−^, N^−^,N_Im_] coordination modes [8,13].

In this manuscript, we report the results obtained in the systematic studies of copper(II), nickel(II) and zinc(II) complexes of multihistidine oligopeptides containing aspartic acid, phenylalanyl and lysyl residues in the sequences. Our aim was the collection of equilibrium, spectroscopic and electrochemical data of multihistidine peptides containing different side chain donor groups and their comparison with those of previously studied multihistidine peptides. These data may provide an overview of the metal binding ability of the –HXH– and –HXZH– motifs, and the fine-tuning effect of the side chain carboxylate and phenylalanine group, which may contribute to a better understanding of the structure and selectivity of the enzymes with similar metal binding sites (e.g., CuZnSOD enzyme).

## 2. Materials and Methods

### 2.1. Materials

In the case of the investigated oligopeptides (Ac-SarHAH-NH_2_, Ac-HADH-NH_2_, Ac-HDAH-NH_2_, Ac-HAHFH-NH_2_, Ac-HFHAH-NH_2_, Ac-HAHKH-NH_2_, Ac-HKHAH-NH_2_, Ac-HAHDH-NH_2_, Ac-HDHAH-NH_2_, Ac-HGFHVH-NH_2_, Ac-HADHAH-NH_2_, Ac-HFHAHFH-NH_2_, Ac-HDHAHDH-NH_2_), solid phase peptide synthesis was performed using a microwave-assisted Liberty 1 Peptide Synthesizer (CEM, Matthews, NC). Fmoc/tBtu technique and TBTU/HOBt/DIPEA activation strategy were used. The detailed description of the procedure has already been explained in our previous papers [17,18,19]. Chemicals and solvents used for synthetic purposes were obtained from commercial sources in the highest available purity and used without further purification. The Rink Amide AM resin (substitution: 0.70 mmole/eq), all of the N-fluorenylmethoxycarbonyl (Fmoc)-protected amino acids (Fmoc-Ala-OH, Fmoc-Asp(O*t*Bu)-OH (O*t*Bu: 5-tert-butyl), Fmoc-Phe-OH, Fmoc-Gly-OH, Fmoc-His(Trt)-OH (TrT: trityl), Fmoc-Lys(Boc)-OH (Boc: tertbutyloxycarbonyl) Fmoc-Val-OH) and 2-(1-H-benzotriazole-1-yl)-1,1,3,3-tetramethyluronium tetrafluoroborate (TBTU) are Novabiochem (Switzerland) products. N-hydroxybenzotriazole (HOBt·H_2_O), N-methylpyrrolidone (NMP), triisopropylsilane (TIS), 2,2′-(ethylenedioxy)diethanethiol (DODT) and 2-methyl-2-butanol were purchased from Sigma-Aldrich Co., while N,N-diisopropyl-ethylamine (DIPEA) and trifluoroacetic acid (TFA) were Merck Millipore Co. products. Peptide-synthesis grade N,N-dimethylformamide (DMF) and acetic anhydride (Ac_2_O) were bought from VWR International, while piperidine, dichloromethane (DCM), diethyl ether (Et_2_O), acetic acid (96%) (AcOH) and acetonitrile (ACN) from Molar Chemicals Ltd. Concentrations of the peptide stock solutions were determined by potentiometric titrations.

The stock solutions of copper(II) chloride, nickel(II) chloride and zinc(II) chloride were prepared from analytical grade reagents, and their concentrations were checked gravimetrically via the precipitation of oxinates.

### 2.2. HPLC

The purity of the synthesized products was checked by analytical RP-HPLC monitoring at the absorbance 222 nm using a Jasco instrument equipped with a Jasco MD-2010 plus multiwavelength detector. Elution method was set as 0% of solvent B at 0 min, which begins to increase after 1 min up to 25% in 14 min and decreases to 0% again after 9 min. The gradient profile was achieved using solvent A (0.1 *v*/*v*% TFA in water) and solvent B (0.1 *v*/*v*% TFA in ACN) at a flow rate of 1.0 mL/min. Solid phase was a Grace Vydac 218TP C18 chromatographic column (250 × 4.6 mm, 300 Å pore size, 5 μm particle size) in the separation procedure.

### 2.3. Potentiometry

First, 3 mL samples of the ligands and 0.2 M carbonate-free potassium hydroxide solution titrant were used for pH-potentiometric measurements. In order to avoid the side reactions with carbon dioxide and/or oxygen, argon gas was added above the samples during titration in which the concentration of the ligand was 2 mM and KCl was a 0.2 M concentration as a background electrolyte. Interaction with metal ions was studied in samples containing copper(II) chloride, zinc(II) chloride or nickel(II) chloride at 1:2, 1:1, 2:1 and 3:1 metal to ligand ratios. All pH-potentiometric measurements were carried out at 298 K, and an IKA Topolino magnetic stirrer was used to stir the samples. To perform the titrations, a MOL-ACS microburette was used, controlled by a computer, and a Molspin pH-meter equipped with a Metrohm 6.0234.100 combination glass electrode detected the pH data, which were converted into hydrogen ion concentrations. PSEQUAD [20] and SUPERQUAD [21] computational programs enabled the calculation of the protonation constants of the ligands and the stability constants of the metal complexes. Equations (1) and (2) define the equilibrium constants:*pM* + *qH* + *rL* ⇆ M*_p_H_q_L_r_*(1)
(2)$$\beta_{pqr}=\frac{\left[M_{p}H_{q}L_{r}\right]}{{\left[M\right]}^{p}\cdot {\left[H\right]}^{q}\cdot {\left[L\right]}^{r}}$$

### 2.4. Spectroscopic Methods

UV–visible absorption spectra were recorded under the same conditions as the pH-potentiometric measurements in 2.5 mL solutions at different pH values. A Perkin Elmer Lambda 25 spectrophotometer (PerkinElmer, Inc., Waltham, MA, USA) was used, and absorption values were registered in the 250–1100 nm wavelength range for copper(II) and nickel(II)-containing systems at a metal to ligand ratio of 1:2 to 3:1 in a 1.00 cm cuvette.

The circular dichroism spectroscopic measurements were carried out on a Jasco-810 spectropolarimeter using the same ligand and metal concentrations and ratios as described above. CD-spectra were recorded in quartz cells of 1.00 cm path length in the 280–800 nm range and 0.10 cm between 220–300 nm.

^1^H-NMR, ^1^H-^1^H COSY and ^1^H-^1^H TOCSY spectra were recorded on a Bruker AM360 MHz FT-NMR and a Bruker Avance 400 spectrometer (Bruker, Fällanden, Switzerland) at 298 K. The chemical shifts were referenced to internal sodium 3-(trimethylsilyl)-1-propane sulfonate (TSP, δ_TSP_ = 0 ppm), and D_2_O was used as a solvent. DNO_3_ and NaOD were used to set the pH of the samples; however, the pH readings were corrected to the hydrogen ion concentration using the appropriate correlation function [22].

### 2.5. Cyclic Voltammetry

All cyclic voltammetric measurements were carried out in aqueous solutions at slightly acidic or physiological pH (0.20 M KNO_3_ was used as supporting electrolyte). An adequate pH was determined from distribution curves of copper(II) complexes of ligands, which was made by means of the “Medusa program” [23]. The pH of the solutions was measured by 827 pH lab pH-meter (Metrohm) equipped with a 6.0234.100 combined electrode (Metrohm, Metrohm AG, Herisau, Switzerland).

The cyclic voltammograms of the copper(II) complexes were obtained by means of a Basi Epsilon Eclipse instrument, except for the peptides Ac-HADH-NH_2_ and Ac-HDAH-NH_2_, in which case the measurements were carried out by Metrohm VA 746 Trace Analyzer equipped with 747 VA Stand driven by a common PC. The complex solutions were degassed using argon gas. In each case, the voltammogram of the deprotonated ligand was registered, and no peaks were found. The systems were analysed at 25 °C with a three-electrode assembly. During the experiments, a glassy carbon electrode (CHI104) was used as the working electrode, depending on the redox properties of the studied complex. The counter electrode was a platinum electrode (distributed by ALS Co. Japan), while the reference electrode was a Vycor tip Ag/AgCl electrode stored in 3 M NaCl (BASI Instr. RE-5B, MF-2079). The concentration of copper(II) ion ranged from 1 to 0.1 mM. The volume of the samples was 1 mL.

The electrochemical measuring system was calibrated with the [Fe(CN)_6_]^3−^/[Fe(CN)_6_]^4−^ redox system. The redox potential was 0.452 V, which is in good agreement with the published redox potential (0.458 V [24]). The potential range changed between +800 and −800 mV. The voltammograms were recorded at 25, 50, 100 and 200 mV/s sweep rate. For the analysis of the voltammograms, we used the eL-Chem Viewer program.

The half-wave potential (*E*_1/2_) values were calculated based on the following equation:(3)$$E_{1/2}=\frac{E_{pc}+E_{pa}}{2}$$
where *E_pc_* and *E_pa_* are the cathodic and anodic peak potentials, respectively.

The half-wave potential value was converted by taking into account that *E*(Ag/AgCl) versus *E*° (vs. NHE) is +209 mV at 25 °C.
*E*° = *E*_1/2_ + *E*°(Ag/AgCl)(4)

Unless otherwise stated, all the potentials (*E*°) reported are referred to NHE.

## 3. Results and Discussion

### 3.1. The Acid-Base Properties of the Peptides

The p*K* values of the ligands completed with the literature data are collected in Table A1 in the Appendix A. The lowest deprotonation constants belong to the carboxyl group of the aspartic acid side chain, the increase in pH results in the deprotonation of the imidazolium groups of histidines, while the highest p*K* values characterize the deprotonation of the ammonium group of the lysine side chain. The p*K*(His) values are close to each other, indicating the overlapping steps of deprotonation of the imidazolium groups. The presence of aspartic acid in the peptide chain leads to a small increase in p*K* values compared to analogous peptides containing the same number and position of histidines, but without other coordinating side chains.

### 3.2. Cu(II) Complexes

The stoichiometry and stability constants of complexes were determined on the base of potentiometric titrations of samples at different copper(II) ion to ligand ratios. These data and the derived constants are collected in Table 1 and Table 2. (The complexes of different ligands with the same stoichiometry have different charges; therefore, the charges are not shown.) The imidazole-coordinated complexes are present in the pH range 4–7. The stoichiometry results of these complexes are different depending on the number of histidines and other donor groups in the peptides. The [CuH_x_L] complexes of three and four histidine-containing peptides correspond the one and two imidazole-coordinated monocomplexes, while the other histidine imidazole groups are protonated. The exceptions are the lysine-containing pentapeptides, where the presence of the lysine ammonium group results in one more proton in the stoichiometry of the complexes. Thus, a direct comparison of the stability constants is not possible, but the derived equilibrium constants can be calculated for the coordination of the ligands through one, two, three and four imidazole nitrogens:M + H_x_L ⇆ MH_x_L(5)
log *K*(M + z(N_Im_)) = log *β*(MH_x_L) − log *β* (H_x_L)(6)
where x is number of the non-coordinated histidine and/or lysine groups. These calculated data can be seen in columns 2–4 of Table 2 and Figure 1.

These data clearly show that the stability of complexes increases in parallel with the increasing number of coordinated imidazoles. The data for one-, two-, three- or four-imidazole-coordinated complexes are more or less similar. However, the presence of a large hydrophobic side chain in the phenylalanine-containing peptides slightly reduces the stability of imidazole-coordinated species, while the aspartic acid side chain slightly increases it due to a weak interaction of the carboxylate group(s). These effects can be illustrated by the distribution of imidazole- and amide-coordinated species of different peptides (Figure 2.)

Similar to the previously studied multhistidine peptides, the formation of imidazole-bound complexes cannot prevent the deprotonation and coordination of amide nitrogens of the peptide backbone. In all cases, the imidazole group of histidine residues is the anchor group that induces the deprotonation and coordination of preceding peptide amide nitrogens. By increasing the pH, the deprotonation and coordination of one-, two- and three amide groups take place, resulting in the formation of [CuH_−1_L], [CuH_−2_L] and [CuH_−3_L] (in the case of a lysine-containing peptide, [CuL], [CuH_−1_L], [CuH_−2_L]) complexes. The deprotonation constants of these processes can be found in the last three columns of Table 2.

These data show that the deprotonation of first and second peptide nitrogens occurs at the lowest pH in the case of peptides containing phenylalanine. It can be explained by the lower stability of imidazole-coordinated complexes, which prevents less the breakage of macrochelates formed by the binding of imidazole rings. Conversely, the stacking interaction between metal ions and aromatic rings, to a small extent, promotes ionization of the amide group.

This effect of aromatic rings was previously observed, e.g., for copper(II) complexes of the phenylalanine-containing dipeptides [25]. The main coordination modes of the complexes are shown in Figure 3.

The spectroscopic parameters (Table A2 in the Appendix A) confirm the structure of the complexes. In the case of the [CuH_−2_L] complex of peptides containing the –HXH– sequence, [N_Im_,N^−^,N^−^,N_Im_] coordination is assumed (Figure 3b). Accordingly, the expected characteristic λ_max_ value based on the empirical formula [26] is 540 nm (assuming that the donor atoms are in the same plane). The measured λ_max_ values for these peptides are generally between 530–560 nm, which support the coordination mode. However, a significantly higher absorption maximum belongs to the [CuH_−2_L] complex of Ac-(HX)*_n_*H-NH_2_ peptides containing one or more aspartic acids, suggesting an axial coordination of the carboxylate group. The λ_max_ value obtained for the [CuH_−2_L] complex of peptides with the Ac-HXYH-NH_2_ sequence is higher than expected for [N^−^,N^−^,N_Im_] coordination (582 nm according to the empirical formula), which supports the axial coordination of the N-terminal histidine and/or carboxylate group. For peptides with the Ac-HXYHZH-NH_2_ sequence, the [CuH_−2_L] complex can be characterized by at least two coordination modes, the ratio of which can be more accurately determined from the CD data (see later).

The [CuH_−3_L] complex is generally characterized by [N^−^,N^−^,N^−^,N_Im_] coordination mode (Figure 3c), which is also reinforced by spectral data (expected λ_max_ value is 522 nm). However, especially for aspartic acid-containing peptides, a small shift in λ_max_ is observed at higher wavelengths.

Further conclusions could be drawn from CD spectroscopic data. In the next section, the effects of side chains are shown in groups of peptides with different length.

#### 3.2.1. Two-Histidine-Containing Tetrapeptides

The complex formation processes differ slightly for Cu(II)-Ac-HADH-NH_2_ and Cu(II)-Ac-HDAH-NH_2_ systems in the pH range 5–11, which is well demonstrated by the titration curves of the two systems (Figure 4a) and the CD spectra recorded around pH 8 (Figure 4b). It can be concluded that the interaction of the carboxylate group is significant in the [CuH_−1_L] complex of the Ac-HADH-NH_2_ peptide and in the [CuH_−2_L] complex of the Ac-HDAH-NH_2_ peptide. These effects are in agreement with the p*K*(amide)_2_ and p*K*(amide)_3_ values of the two tetrapeptides; p*K*(amide)_2_ is higher for Ac-HADH-NH_2_, while p*K*(amide)_3_ is higher for the Ac-HDAH-NH_2_ peptide, confirming that the coordinated carboxylate group hinders the deprotonation of the next amide nitrogen (see Table 2 and Appendix A, Figure A1). The carboxylate group, however, has no effect on the structure of [CuH_−3_L] complexes, the CD spectra are similar to each other and to that of Ac-HAAH-NH_2_ (Appendix A, Figure A2).

#### 3.2.2. Three-Histidine-Containing Pentapeptides

In the complexes of Ac-HXHZH-NH_2_ peptides coordinated by the [N_Im_,N^−^,N^−^,N_Im_] or [N^−^,N^−^,N^−^,N_Im_] donor set, the metal ion may bind to either the C-terminal histidine or the intermediate histidine, and thus, coordination isomers can exist. The results of previous studies have shown that the metal binding at the C-terminus is more favourable, but if the amino acid between the two histidines has a sterically large side chain, the ratio of complexes containing the metal ion bound to intermediate histidine increases.

Based on the spectral data of the [CuH_−2_L] complexes of different peptides, we obtained similar CD spectra for the phenylalanine- and lysine-containing pentapeptides regardless of the sequence as the previously studied valine- and alanine-containing pentapeptides. There is a difference in the [CuH_−2_L] complex of the two aspartic acid-containing peptides, where the carboxylate coordination results in a significant difference in the CD spectrum (Figure 5a). This effect is independent of whether the aspartic acid is in the C-terminal or N-terminal part of the molecule. However, this difference was not observed for [CuH_−3_L] complexes of aspartic acid-containing pentapeptides (Figure 5b).

Conversely, the CD spectra of the [CuH_−3_L] complex of the Ac-HAHFH-NH_2_ peptide (light brown curve in Figure 5b) differ from the others. This suggests that in this case the formation of that complex is preferred, in which the metal ion is coordinated by the C-terminal –HFH– sequence, which means that the stacking interaction between the aromatic ring of the phenylalanine and the Cu(II) ion favours the binding to this sequence despite the presence of the sterically large side chain. The results obtained for the Ac-HFHAHFH-NH_2_ peptide are in agreement with this conclusion.

The presence of more than one histidine in the molecule also allows for the binding of more than one metal ion and the formation of dinuclear complexes. For peptides with the Ac-HXHAH-NH_2_ sequence (X = F, D or K), the formation of dinuclear complexes was observed in all cases, while for the Ac-HAHDH-NH_2_ peptide, the excess of metal ions led to precipitation. The formation of dinuclear complexes is supported by CD spectroscopic measurements, where the spectra recorded in systems at Cu(II)-L = 2:1 ratio differ from that of a 1:1 ratio. This is demonstrated by Figure A3 (in the Appendix A), where the spectra of the corresponding copper(II) complexes of Ac-HDHAH-NH_2_, Ac-HKHAH-NH_2_ and Ac-HFHAH-NH_2_ are shown.

#### 3.2.3. Three-Histidine-Containing Hexapeptides

For the Ac-HXXHZH-NH_2_ peptides, the binding of metal ions to the C-terminal histidine and the intermediate histidine results in different binding modes. Thus, the CD studies offer an estimation of the ratio of the two coordination isomers (Figure 6, Table 3).

In the case of Ac-HGFHVH-NH_2_, the ratio of isomers was estimated using the CD spectra of Ac-HGGH-NH_2_ and Ac-HVH-NH_2_, while in the case of Ac-HADHAH-NH_2_, the CD spectra of Ac-HADH-NH_2_ and Ac-HVH-NH_2_ were used. In both cases, due to the different sequence of model peptides, only the approximate ratio could be determined.

The data collected in Table 3 reflect well that the presence of aspartic acid and phenylalanine preceding the intermediate histidine increases the ratio of complexes in which the metal ion binds to the intermediate part of the molecule. Similar to the pentapeptides, the stacking interaction of the aromatic side chain of phenylalanine and axial interaction of the carboxylate group of aspartic acid makes this coordination mode favoured. In the case of the [CuH_−3_L] complex, the ratio of metal binding to the intermediate histidine increases further, similarly to the Ac-HAAHVH-NH_2_ peptide. It can be generally stated that the –HXZH– sequence offers a sterically more favourable position for the [N^−^,N^−^,N^−^,N_Im_] coordination mode than for the –HXH– sequence at the C-terminal part.

#### 3.2.4. Four-Histidine-Containing Heptapeptides

For the heptapeptides Ac-HDHAHDH-NH_2_ and Ac-HFHAHFH-NH_2_, mono-, di- and tri-nuclear complexes were formed depending on the ratio of the copper(II) ligand. Histidines as independent anchor groups can bind up to three copper (II) ions. As the ratio of metal ion to ligand is increased from 1:1 to 2:1 and 3:1, multinuclear (di- and tri-nuclear) complexes dominate in the alkali pH range, which is followed by spectral changes as well (see Figure 7, Figure A4 and Figure A5 in Appendix A). For the dinuclear complexes, the formation of coordination isomers can also be expected. For both heptapeptides, the CD spectra of the [Cu_2_H_−4_L] complexes formed around pH 9 are similar to those of the aspartic acid- and phenylalanine-containing Ac-HXHZH-NH_2_ pentapeptides (Figure A5a,c in Appendix A). However, the spectra of the [Cu_2_H_−5_L] complex formed in the strongly alkaline medium are similar to those of Ac-HAHDH-NH_2_ and Ac-HAHFH-NH_2_, respectively, and differ from those of the dinuclear complexes of the aforementioned pentapeptides (Figure A5b,d). This confirms the conclusion that for both metal ions the –HDH– and –HFH– sequences (on the C- and N-terminus) are the major binding sites.

#### 3.2.5. Electrochemical Properties of the Copper(II) Complexes

Since the side chain effect of aspartic acid was the most significant in copper(II) complexes of multihistidine peptides, the cyclic voltammetric measurements of copper(II) complexes of these peptides were also performed to investigate whether the aspartic acid side chain had an effect on the redox properties of the copper(II) complexes.

The electrochemical parameters of the species with different coordination modes are shown in Table 4 and Table A3. As these data and Figure 8 demonstrate, CV curves of 2 × N_Im_ and 4 × N_Im_ ([CuL] and [CuL_2_]) and amide nitrogens-coordinated complexes ([CuH_−1_L] and [CuH_−2_L]) were registered in the measurable potential range. The *E*° (vs. NHE) values of 2 × N_Im_ and 3 × N_Im_ coordinated species follow the expected trend [25], namely that the increase in the number of imidazole nitrogens in the coordination sphere and the stability of [CuL] complexes is accompanied by a decrease in redox potential values (Figure 9a).

The data also reflect that the redox potential values (vs. NHE) of the imidazole-coordinated complexes are similar to those of the previously examined peptides containing an aliphatic side chain [13,27]. Thus, the redox properties of imidazole-coordinated complexes are practically not affected by the presence of aspartic acid.

However, a more significant difference can be observed in the case of copper(II) complexes coordinated by the [N_Im_,N^−^,N^−^,N_Im_] donor set (Figure 9b). The redox potential of the [CuH_−2_L] complex of aspartic acid-containing peptides is increased compared to the previously studied peptides.

Studies of Cu(II) complexes with different type of ligands have shown that the rigid structure of the Cu(II) complex cannot satisfy the coordination requirement of Cu(I). This explains that copper(II) complexes with quasi-square planar geometry ([NH_2_,N^−^,N^−^,N^−^] coordinated species of oligopeptides, [NH_2_,N^−^,N^−^,N_Im_] coordinated complex of GGH peptide, [N^−^,N^−^,N^−^,N_Im_] coordinated complexes of terminally protected histidine-containing peptides) have low (negative) redox potential values. The reduction process of Cu(II) complexes has been proved to be facilitated when the Cu species can easily evolve towards a tetrahedral geometry upon reduction, which increases the stability of Cu(I) complex [28,29,30,31,32,33].

Based on these facts, the redox potential values fall in the positive range due to the distorted geometry of the [CuH_−2_L] complex of multihistidine peptides, and the geometry of the complex is further distorted by the interaction between the metal centre and the carboxylate group of aspartic acid.

### 3.3. Ni(II) Complexes

Nickel(II) complexes of the aforementioned multihistidine peptides were also studied. These measurements were completed by the characterization of three other multihistidine peptides (Ac-HAAH-NH_2_, Ac-HAAHVH-NH_2_ and Ac-HAAHGH-NH_2_), whose copper(II) complexes had been previously investigated [13]. The stoichiometry and stability constants of the complexes are collected in Table 5 and Table 6.

Complex formation processes can be observed above pH 4. Imidazole-coordinated nickel(II) complexes exist over a wide pH range (pH 4–8), including the physiological pH range.

The derived formation constants related to coordination of imidazole nitrogens reflect that, similar to the copper(II) ion, the stability of the imidazole-coordinated nickel(II) complexes decreases when a large hydrophobic phenylalanyl side chain close to the N-terminal part or a lysyl side chain is present in the molecule, while the carboxylate group slightly increases the stability of the complexes.

The metal ion-induced deprotonation and coordination of one, two and three peptide nitrogens take place above pH 8. Typically, the cooperative deprotonation of first and second amide nitrogens takes place, while the third deprotonation step is observed at about a one-pH unit larger pH. However, in the case of the Ac-HAAH-NH_2_, Ac-HDAH-NH_2_ and Ac-HGFHVH-NH_2_ peptides, the [NiL] complex is converted to the [NiH_−3_L] complex in practically one step, and the [N^−^,N^−^,N^−^,N_Im_] coordinated species is predominant above pH 9. This suggests that nickel(II) binding to the –HXZH– sequence is preferred in these cases. For the Ac-HADH-NH_2_ peptide in addition to the coordination of N^−^ and N_Im_ donor groups, the participation of the carboxylate group is also assumed, resulting in a (7,5) joined chelate ring, which shifts the deprotonation of the second and third amide nitrogens into the higher pH range.

Spectroscopic data support these coordination modes, but the preferred metal binding site for molecules can only be concluded in a few cases. The imidazole-coordinated complexes have an octahedral geometry supported by characteristic absorption bands with low intensity (Table A4). These complexes are not CD active. The deprotonation of amide nitrogens is accompanied by the appearance of an intensive band around 435 nm, and the intensity of this band increases in parallel with the formation of [NiH_−2_L] and [NiH_−3_L], which corresponds to the deprotonation of two and three amide nitrogens (Figure 10).

In the case of tetrapeptides containing two histidines, the main anchor group is the C-terminal histidine imidazole nitrogen, and the deprotonation of preceding amide nitrogens takes place in the alkali media. The CD spectra of the peptides containing terminal histidines are similar, and a smaller difference can be observed for the Ac-SarHAH-NH_2_ peptide due to the different position of histidines (Figure A6 in Appendix A). Based on these data, it can also be concluded that the −HADH− sequence is the main binding site for the Ac-HADHAH-NH_2_ peptide, since the CD spectrum is similar to that of the Ac-HADH-NH_2_ peptide and differs from those of both the Ac-HAHVH-NH_2_ and Ac-SarHAH-NH_2_ peptides (Figure 11a).

A similar conclusion can be drawn for the Ni(II) complexes of the other Ac-HXZHXH-NH_2_ peptides. The CD spectrum of Ac-HGFHVH-NH_2_ provides clear evidence for this finding (Figure 11b). This spectrum differs from that of the other peptides, with cotton effects in the d-d range having opposite signs. This confirms that Ni(II) binds to the –HGFH– moiety, as it is known that binding to the GZH sequence results in the opposite cotton effect as the binding to XZH sequence (where X is an amino acid other than glycine) [34]. This also means that this binding mode is favourable in the presence of an aromatic phenyl ring. Moreover, the presence of phenylalanyl residue promotes the formation of a [NiH_−3_L] complex with coordination of a [N,N^−^,N^−^,N_Im_] donor set compared to other peptides.

Ac-HXHZH-NH_2_ peptides behave similarly to the previously studied peptides. In these cases, it was concluded that if a large side chain is present in the C-terminal part of the molecule, the ratio of complexes in which the metal ion coordinates to the C-terminal and intermediate histidine is comparable. This results in significantly different CD spectra, especially for the [NiH_−2_L] complex (Figure 12).

This difference in the CD spectrum is observed for the [NiH_−2_L] complex of peptides containing aspartic acid and phenylalanine at the C-terminal position, which means that both aspartic acid and phenylalanyl side chains are stabilizing factors in the complexes of peptides containing –HXZH– sequence, whereas for the –HDH– and –HFH– sequences, the presence of both aspartic acid and phenylalanine side chains hinders sterically the formation of amide nitrogen coordinated complexes.

### 3.4. Zn(II) Complexes

Since the studies of copper(II) and nickel(II) complexes have shown that only the presence of phenylalanine and aspartic acid had a more significant effect on the complex formation processes, only the investigation of zinc(II) complexes of these two types of peptides was planned. However, complexes of phenylalanine-containing peptides were practically impossible to characterize due to precipitation in the whole physiological and alkali pH range.

The stability constants of zinc(II) complexes of aspartic acid-containing peptides are collected in Table 7. The ligand is coordinated via imidazole nitrogens to the zinc(II) ion in the [ZnHL] and [ZnL] complexes. The formation constants characteristic of imidazole-coordinated complexes are slightly larger than those of the previously studied peptides (for Ac-HVVH-NH_2_ peptide: log *β* (ML) = 3.57, [10] Ac-HVHAH-NH_2_: log *β* (ML) = 5.09 [35]). The interaction of the carboxylate group results in an increase in the stability of these complexes. By raising the pH, generally, either precipitation or the formation of mixed hydroxido complexes can be assumed, as in the case of the previously studied Ac-HXHZH-NH_2_ peptides.

However, a different process was observed for peptides containing the –HADH– sequence, where no precipitation was observed in the strongly alkaline range either. Thus, deprotonation of the amide nitrogens was assumed for the [ZnH_−1_L] and [ZnH_−2_L] complexes formed in parallel with the extra alkali consumption process (Figure 13).

The zinc(II)-induced amide nitrogen deprotonation processes were already published for some histidine-containing peptides previously. Similar interactions were observed in the case of the Ac-HHVGD-NH_2_ peptide [35], and deprotonation of one and two amide nitrogen was also described in the [ZnH_−1_L] and [ZnH_−2_L] complexes of Ac-GHEITHG-NH_2_ and Ac-GHTIEHG-NH_2_ peptides with HExxH and HxxEH motifs [36].

^1^H NMR spectroscopy was performed to confirm the assumed structure of the [ZnH_−1_L] and [ZnH_−2_L] complexes of Ac-HADH-NH_2_. ^1^H NMR spectra of the peptide and Zn(II):Ac-HADH-NH_2_ = 1:1 at similar pH values were registered. Two-dimensional spectra were also recorded to facilitate signal assignment and to analyse the spectral differences between the complexes formed at different pH.

The signals of the imidazole H_δ_ and H_ε_ protons of histidines in the aromatic range were monitored by ^1^H-NMR spectra. The pH-dependent ^1^H-NMR spectra and assignment of signals can be seen in Figure A7 and Figure A8 in Appendix A. The spectral studies confirmed that the [ZnL] complex formed in the physiological pH range is coordinated by both histidine imidazole nitrogens. With increasing pH, in addition to the signal of the coordinated imidazole protons in the Zn(II) complex, the signals of the free peptide also appeared. (Figure A8). This means that above pH 8, one of the imidazole nitrogen atoms is replaced with an amide nitrogen donor in the [ZnH_−1_L] complex. To prove this, ^1^H-^1^H NMR spectra of the peptide and Zn(II) complexes were evaluated in the alkaline region, and the signals are summarized in Table A5 in Appendix A. A comparison of ^1^H-^1^H NMR spectra of the peptide and the Zn(II) complex at pH~8.2 is shown in Figure 14, while a comparison of the ^1^H-^1^H NMR spectra of Zn(II) complexes at pH~8.2 and pH~10.8 can be seen in Figure 15.

As can be seen from Table A5 and Figure 14, the deprotonation of amide nitrogens can be concluded from the signs of the methine group of histidines and aspartic acid. If the amide nitrogen preceding the C-terminal histidine residue is deprotonated and coordinated, the signal of α-CH of the histidine is shifted towards a higher field strength, because this proton is sensitive to the negative charge (at pH~8.2 His α-CH 4.56 ppm for the free ligand and 3.94 ppm for the complex).

By increasing the pH to around 11, the signal of the methine proton of histidine hardly changes. However, in the case of α-CH of aspartic acid, another signal could be identified by comparing the TOCSY spectra of Zn(II) complexes at pH 8 and pH 11 (at pH~10.8, Asp α-CH 4.55 ppm for the free ligand and 3.53 ppm for the complex (Table A5, Figure 15)). As a conclusion, above pH 8, the second amide nitrogen was deprotonated and coordinated. Figure 15 shows the spectrum of the complex [ZnH_−1_L] recorded at pH 8.20 (red) and the spectrum of the complex [ZnH_−2_L] recorded at pH 10.8 together (black).

## 4. Conclusions

Systematic studies of terminally protected peptides containing two, three, and four histidines mimicking the sequence of the metal binding site of CuZnSOD (Ac-SarHAH-NH_2_, Ac-HADH-NH_2_, Ac-HDAH-NH_2_, Ac-HXHYH-NH_2_ X, Y = A, F, D or K, Ac-HXHAHXH-NH_2_, X = F or D) were performed. The results obtained for copper(II), nickel(II) and zinc(II) complexes have shown how the side chains of different amino acids (aspartic acid, phenylalanine, lysine) affect the complex formation processes and the electrochemical parameters of copper(II) complexes. In addition, the results provided an opportunity to draw comparisons with those of previously studied peptides with similar sequences and to summarize the equilibrium, spectroscopic and electrochemical parameters in summary tables.

It has been found that the aspartic acid side chain has the greatest effect. The presence of the negatively charged side chain and the weak axial coordination of the carboxylate group increases the stability of the imidazole-coordinated complexes formed with all three studied metal ions. Coordination of the carboxylate group can also be assumed in amide nitrogen-coordinated Cu(II) complexes. This results in an increase in the redox potential values of the [CuH_−1_L] and [CuH_−2_L] complexes and a higher reducibility of these complexes due to a greater degree of distorted geometry. In addition, the presence of aspartic acid has a significant effect on the complex formation processes that take place in the presence of zinc(II). In the case of peptides containing the −HADH− and −HDAH− sequences, the coordination of the carboxylate group stabilizes the zinc(II) complexes in the physiological pH range and promotes the zinc(II) ion-induced amide nitrogen deprotonation and coordination.

A smaller effect of the phenylalanine side chain was observed. The stacking interaction between the aromatic ring and the metal ion increases the stability of the complexes. However, due to the presence of large hydrophobic side chains, the solubility of the complexes significantly decreases, which made the characterization of these complexes difficult and did not allow for the study of zinc(II) complexes.

However, for both aspartic acid and phenylalanine-containing peptides, coordination isomers of [CuH_−x_L] (x = 1–3) complexes are formed, the ratios of which are influenced by the position of aspartic acid or phenylalanine amino acids. The stacking interaction of aromatic side chains of phenylalanine and axial interactions of carboxylate group of aspartic acid make the binding of copper(II) ion to the intermediate histidine favoured in the case of Ac-HXZHXH-NH_2_ (Z = D or F) hexapeptides, while −HFH− or −HDH− sequences are the preferred metal ion binding sites for pentapeptides. Based on these results, we continue our work with synthesis and the study of 10–15-membered peptides containing the copper(II) and zinc(II) binding sites of the CuZnSOD enzyme together.

## Figures and Tables

**Figure 1 molecules-27-03435-f001:**
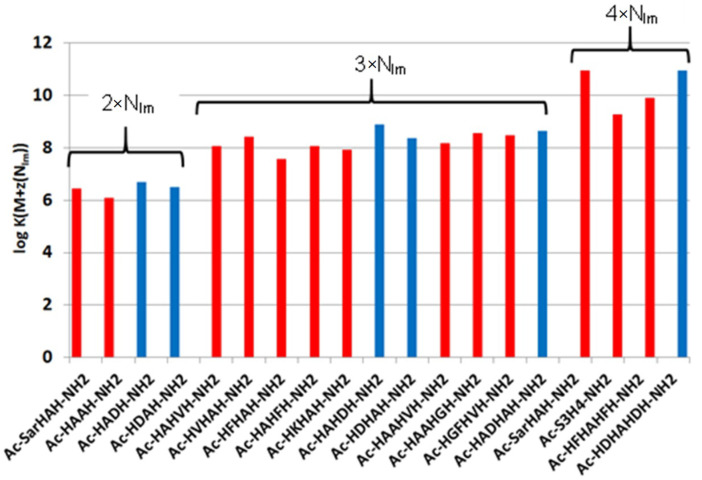
Equilibrium constants of imidazole-coordinated copper(II) complexes (log *K*(M + z(N_Im_))) of various histidine-containing peptides (data from refs. [8,9,13]); (■) data-related complexes of peptides containing aspartic acid.

**Figure 2 molecules-27-03435-f002:**
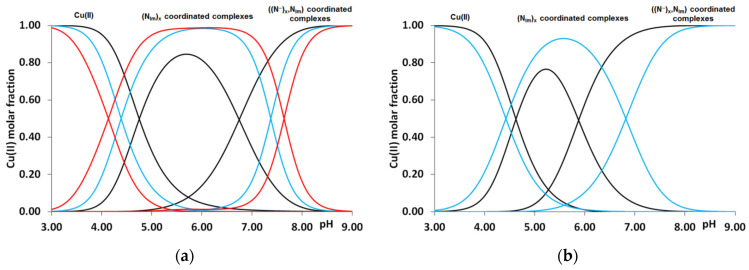
Distribution of imidazole- and amide nitrogen-coordinated complexes formed in equimolar solution of Cu(II)-Ac-HADH-NH_2_ (■), Cu(II)-Ac-HADHAH-NH_2_ (■) and Cu(II)-Ac-HDHAHDH-NH_2_ (■) (**a**). Cu(II)-Ac-HFHAH-NH_2_ (■) and Cu(II)-Ac-HDHAH-NH_2_ (■) (c_L_ = c_M_ = 2 mM) (**b**).

**Figure 3 molecules-27-03435-f003:**
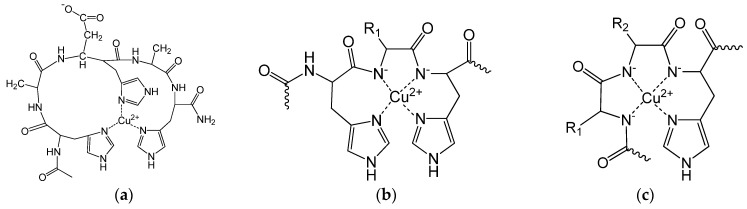
Schematic structure of the main complexes: [CuL], L = Ac-HADHAH-NH_2_ (**a**), [CuH_−2_L], L = Ac-(HX)*_n_*H-NH_2_ (**b**), [CuH_−3_L] (**c**).

**Figure 4 molecules-27-03435-f004:**
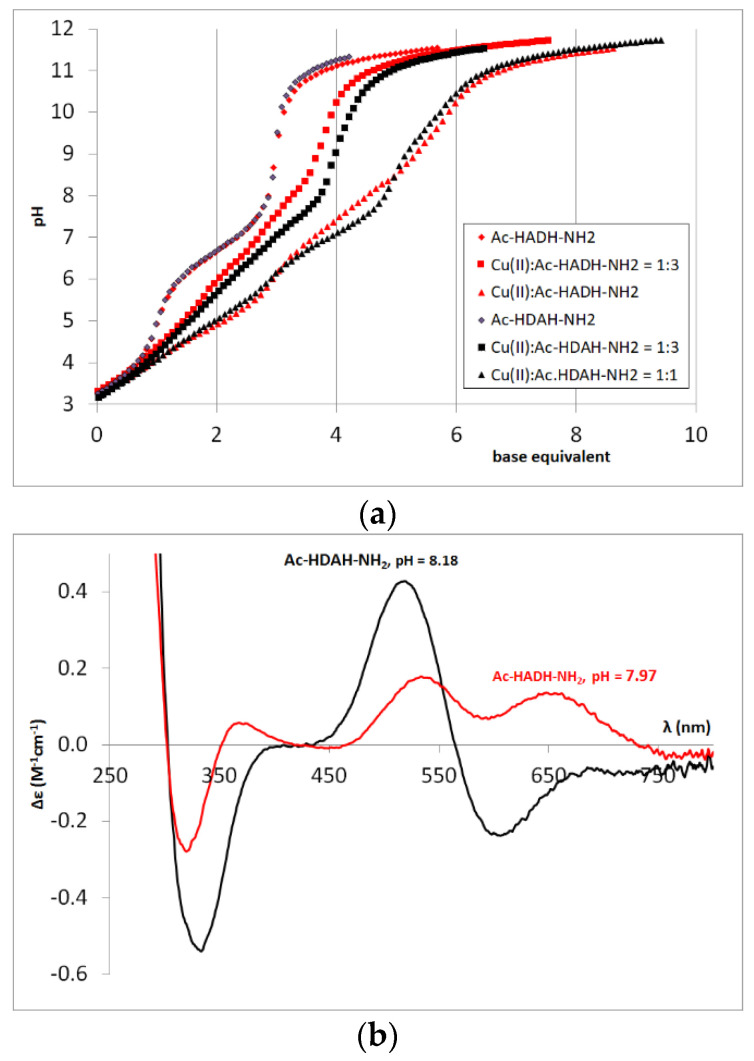
Titration curves of L, Cu(II)-L = 1:3 and 1:1 systems (L = Ac-HADH-NH_2_, Ac-HDAH-NH_2_) (**a**); CD spectra registered in equimolar solution of Ac-HADH-NH_2_ (■) and Ac-HDAH-NH_2_ (■) systems at pH 8 (c_L_ = 2 mM) (**b**).

**Figure 5 molecules-27-03435-f005:**
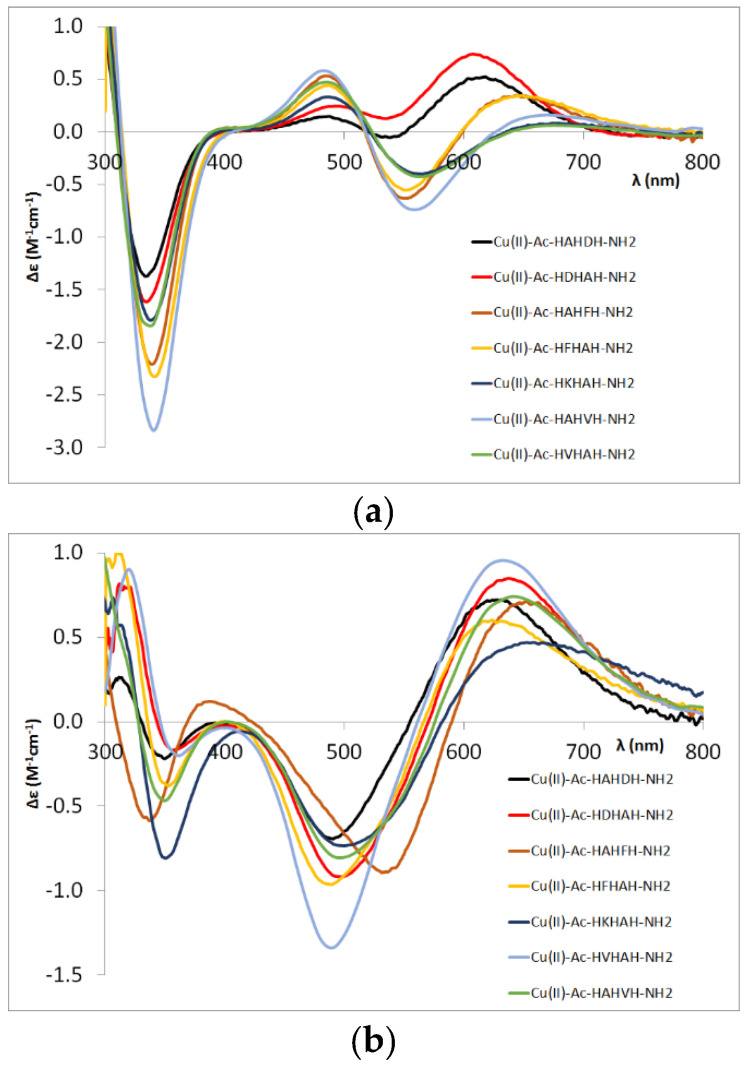
CD spectra of the [CuH_−2_L] (**a**) and [CuH_−3_L] (**b**) complexes of different pentapeptides.

**Figure 6 molecules-27-03435-f006:**
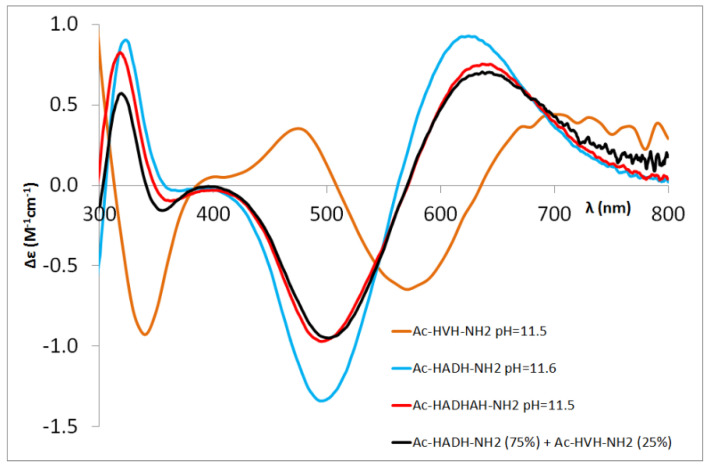
Comparison of experimental CD spectra of the [CuH_−3_L] complex of Ac-HADH-NH_2_ (blue), Ac-HVH-NH_2_ (orange) [5] and Ac-HADHAH-NH_2_ (red) with superposition of CD spectra of [CuH_−3_L] of Ac-HADH-NH_2_ and Ac-HVH-NH_2_ in 75:25% (black).

**Figure 7 molecules-27-03435-f007:**
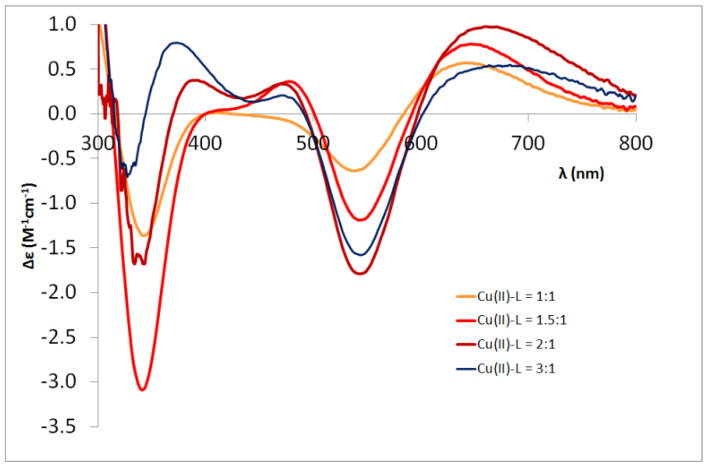
The CD spectra of the Cu(II)-Ac-HFHAHFH-NH_2_ system in function of the Cu(II)-L ratio (pH = 10.5).

**Figure 8 molecules-27-03435-f008:**
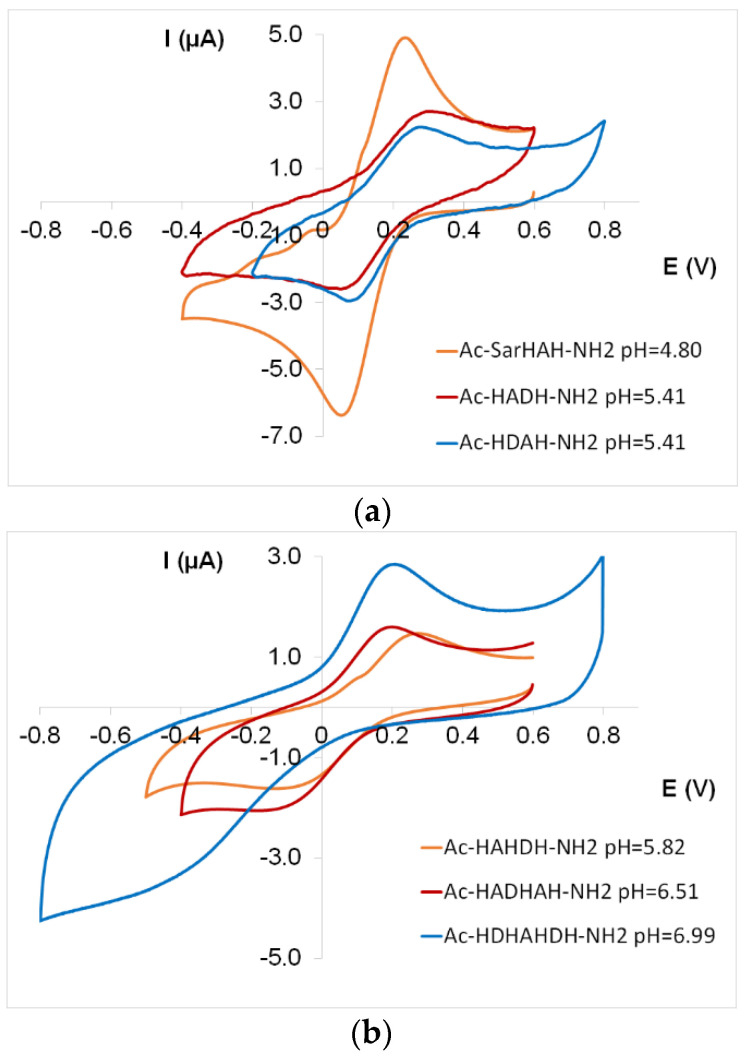
Cyclic voltammogram of [CuL] complex of two histidine-containing tetrapeptides (Ac-SarHAH-NH_2_, Ac-HADH-NH_2_, Ac-HDAH-NH_2_) (**a**) and three histidine-containing penta-, hexa- and heptapeptides (Ac-HAHDH-NH_2_, Ac-HADHAH-NH_2_, Ac-HDHAHDH-NH_2_) (**b**).

**Figure 9 molecules-27-03435-f009:**
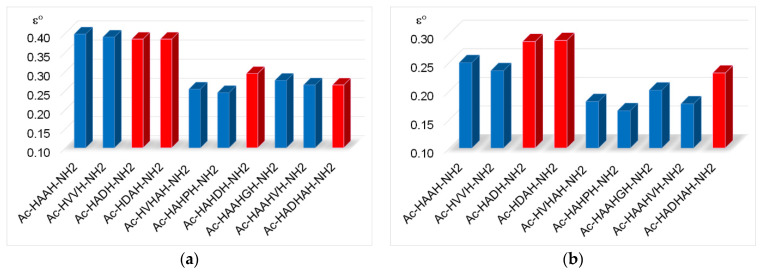
Redox potentials (vs. NHE) of [CuL] (**a**) and [CuH_−2_L] (**b**) complexes of aspartic acid containing (■) and other multihistidine peptides (■) (Data from refs. [13,27]).

**Figure 10 molecules-27-03435-f010:**
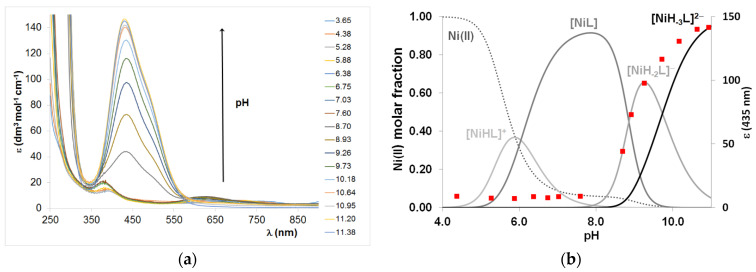
UV–Vis spectra (**a**) and concentration distribution curves of equimolar solution of Ni(II):Ac-HADHAH-NH_2_ system and the absorption values at 435 nm in function of pH (c_L_ = 2 mM) (**b**).

**Figure 11 molecules-27-03435-f011:**
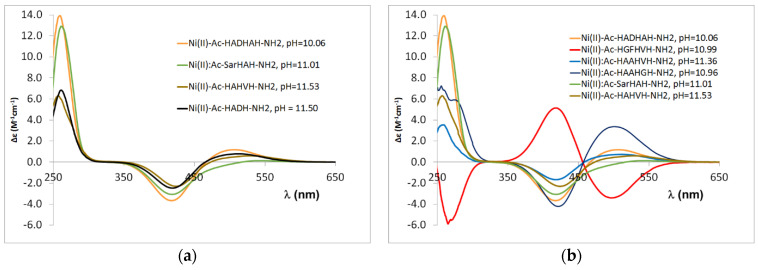
CD spectra registered in equimolar solution of Ni(II)-multihistidine peptide systems at pH 11.0–11.5; data of Ac-HAHVH-NH_2_ from Ref. [9]. (**a**) CD spectra comparision of Ac-HADHAH-NH_2_, Ac-SarHAH-NH_2,_ Ac-HAHVH-NH_2_ and Ac-HADH-NH_2_; (**b**) CD spectra comparision of Ac-HADHAH-NH_2_, Ac-HGFHVH-NH_2_, Ac-HAAHVH-NH_2_, Ac-HAAHGH-NH_2_, Ac-SarHAH-NH_2_, Ac-HAHVH-NH_2_.

**Figure 12 molecules-27-03435-f012:**
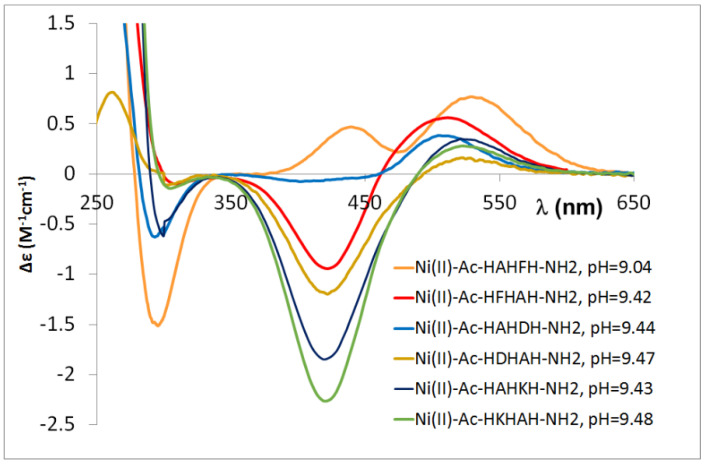
CD spectra registered in equimolar solution of Ni(II)-multihistidine pentapeptide systems at pH 9.0–9.5.

**Figure 13 molecules-27-03435-f013:**
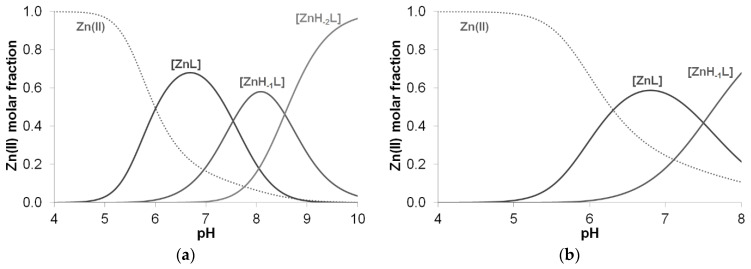
Distribution curves of complexes formed in the equimolar solution of Zn(II)-Ac-HADH-NH_2_ (**a**) and Zn(II)-Ac-HDAH-NH_2_ (**b**) (c_L_ = c_M_ = 2 mM).

**Figure 14 molecules-27-03435-f014:**
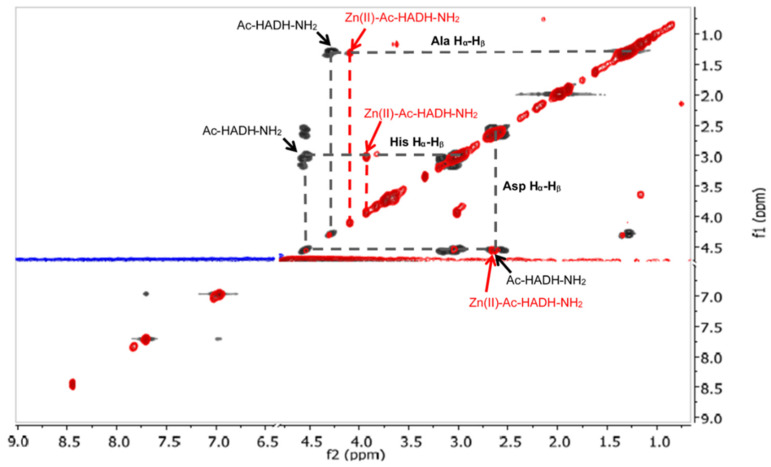
^1^H-^1^H TOCSY spectra of the Zn(II)-Ac-HADH-NH_2_ system (red) compared to the free Ac-HADH-NH_2_ ligand (black) (pH~8.2). The presence of the zinc(II) ion causes different chemical shifts of two histidyl and alanyl methine protons. The marked cross peaks are derived from the correlation between the H_β_ and H_α_ protons.

**Figure 15 molecules-27-03435-f015:**
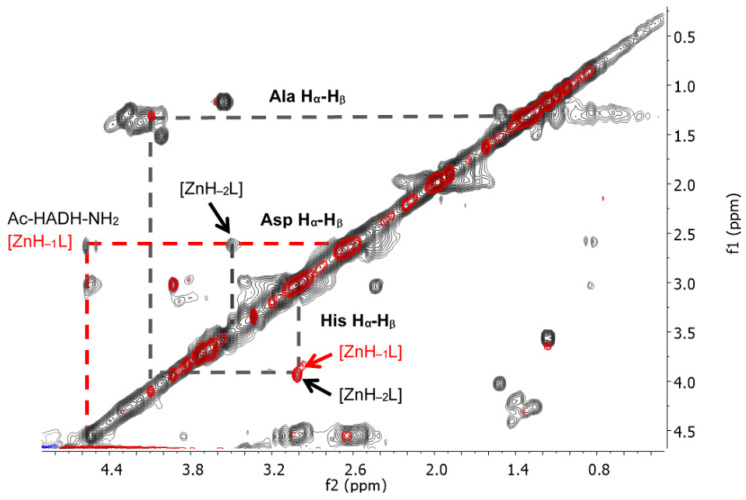
^1^H-^1^H TOCSY spectra of the Zn(II)-Ac-HADH-NH_2_ system at pH~8.2 (red) and at pH~10.8 (black). The increase in the pH causes different chemical shifts of aspartic acid methine protons. The marked cross peaks are derived from the correlation between the H_β_ and H_α_ protons.

**Table 1 molecules-27-03435-t001:** Stability constants (log *β*) of complexes formed in the different Cu(II)-multihistidine peptide systems (T = 298 K, I = 0.2 M KCl).

	Ac-SarHAH-NH_2_	Ac-HADH-NH_2_	Ac-HDAH-NH_2_	Ac-HFHAH-NH_2_	Ac-HAHFH-NH_2_	Ac-HKHAH-NH_2_	Ac-HAHDH-NH_2_	Ac-HDHAH-NH_2_	Ac-HGFHVH-NH_2_	Ac-HADHAH-NH_2_	Ac-HFHAHFH-NH_2_	Ac-HDHAHDH-NH_2_
[CuL_2_]	10.96(5)	–	–	–	–	–	–	–	–	–	–	–
[CuH_3_L]	–	–	–	–	–	–	–	–	–	–	–	25.10(3)
[CuH_2_L]	–	–	–	–	–	23.24(3)	18.28(8)	17.57(6)	–	–	19.39(2)	–
[CuHL]	–	–	–	12.79(3)	13.25(1)	18.22(3)	13.66(6)	13.49(2)	13.28(4)	13.64(2)	14.79(2)	16.54(2)
[CuL]	6.45(2)	6.70(4)	6.50(4)	7.57(4)	8.08(1)	11.70(6)	8.89(4)	8.38(2)	8.49(1)	8.65(2)	9.94(2)	10.95(5)
[CuH_−1_L]	–	−0.07(1)	−0.25(8)	1.76(4)	2.04(1)	4.67(5)	1.98(7)	1.49(3)	1.07(4)	–	3.36 (5)	–
[CuH_−2_L]	−6.21(2)	−8.07(9)	−7.37(6)	−5.77(7)	−4.63(1)	−5.02(8)	−5.23(6)	−6.08(3)	−6.80(6)	−6.20(4)	−4.00(7)	−4.36(6)
[CuH_−3_L]	−15.45(4)	−16.40(9)	−16.56(9)	−16.85(8)	−15.59(1)	−15.35(7)	−15.47(7)	−16.9(1)	−15.89(6)	−15.70(6)	−14.74(2)	−14.61(8)
[Cu_2_L]	–	–	–	–	–	–	–	–	–	–	12.70(4)	–
[Cu_2_H_−1_L]	–	–	–	–	–	8.6(1)	–	–	–	–	–	–
[Cu_2_H_−2_L]	–	–	–	–	−0.44(1)	–	–	–	–	−1.02(6)	2.37(2)	3.26(3)
[Cu_2_H_−3_L]	–	–	–	–	–	−3.99(7)	–	−8.09	–	−7.54(4)	–	–
[Cu_2_H_−4_L]	–	–	–	−13.24(5)	−11.79(1)	–	–	−17.7(3)	–	−14.64(5)	−9.62(3)	−12.16(5)
[Cu_2_H_−5_L]	–	–	–	−23.58(9)	−22.39(1)	−24.8(1)	–	−29.8(3)	–	−23.64(8)	−20.22(5)	−22.59(8)
[Cu_2_H_−6_L]	–	–	–	–	–	–	–	–	–	−34.54(7)	−31.41(3)	−33.84(6)
[Cu_3_H_−5_L]	–	–	–	–	–	–	–	–	–	–	–	−14.6(1)
[Cu_3_H_−6_L]	–	–	–	–	–	–	–	–	–	–	−18.85(3)	−23.7(5)
[Cu_3_H_−7_L]	–	–	–	–	–	–	–	–	–	–	−28.68(4)	−32.9(2)

**Table 2 molecules-27-03435-t002:** Derived equilibrium constants of imidazole-coordinated complexes and deprotonation constants of amide nitrogen coordinated complexes.

	lg*K*(Cu(II) + N_Im_)	lg*K*(Cu(II) + 2N_Im_)	lg*K*(Cu(II) + 3N_Im_)	lg*K*(Cu(II) + 4N_Im_)	p*K*(Amide)_1_	p*K*(Amide)_2_	p*K*(Amide)_3_
Ac-HVH-NH_2_ [6]	–	6.63	–	11.58	6.25 (av.)		10.34
Ac-SarHAH-NH_2_	–	6.45	–	10.96	6.33 (av.)		9.24
Ac-HAAH-NH_2_ [2]	–	6.08	–	–	7.01	7.12	8.34
Ac-HADH-NH_2_	–	6.70	–	–	6.77	8.00	8.33
Ac-HDAH-NH_2_	–	6.50	–	–	6.75	7.12	9.19
Ac-HAHVH-NH_2_ [8]	–	6.25	8.08	–	6.81	7.12	9.96
Ac-HVHAH-NH_2_ [8]	–	6.39	8.42	–	6.59	7.40	9.53
Ac-HFHAH-NH_2_	–	5.76	7.57	–	5.81	7.53	11.08
Ac-HAHFH-NH_2_	–	6.30	8.08	–	6.04	6.67	10.96
Ac-HKHAH-NH_2_	–	6.12	7.94	–	6.52	7.03	9.69
Ac-HAHDH-NH_2_	4.61	6.48	8.89	–	6.91	7.21	10.24
Ac-HDHAH-NH_2_	3.96	6.35	8.38	–	6.89	7.57	10.81
Ac-HAAHVH-NH_2_ [13]	–	6.22	8.17	–	7.04	7.97	9.40
Ac-HAAHGH-NH_2_ [13]	–	6.02	8.55	–	7.02	7.43	9.05
Ac-HGFHVH-NH_2_	–	6.32	8.49	–	7.42	7.87	9.80
Ac-HADHAH-NH_2_	–	6.53	8.65	–	7.42 (av.)		9.50
Ac-S3H4-NH_2_ * [9]	–	5.89	7.41	9.29	–	–	–
Ac-HFHAHFH-NH_2_	–	6.16	7.99	9.94	6.58	7.36	10.74
Ac-HDHAHDH-NH_2_	4.89	–	9.37	10.95	7.65 (av.)		10.25

* Ac-S3H4-NH_2_ is the abbreviation of Ac-His-Sar-His-Sar-His-Sar-His-NH_2_ peptide.

**Table 3 molecules-27-03435-t003:** The estimated ratio of –HXZH– coordinated and –HXH– coordinated species of three-histidine-containing hexapeptides.

	Ac-HGFHVH-NH_2_	Ac-HADHAH-NH_2_	Ac-HAAHVH-NH_2_ [11]
**[CuH_−2_L]**	50:50	60:40	10:90
**[CuH_−3_L]**	60:40	75:25	80:20

**Table 4 molecules-27-03435-t004:** Redox potentials (vs. NHE) of the Cu(II) complexes of ligands.

Coordination Mode	2 × N_Im_	3 × N_Im_	4 × N_Im_	[N_Im_,N^−^,N_Im_]	[N_Im_,N^−^,N^−^,N_Im_]
	[CuL]		[CuL_2_]		
Ac-SarHAH-NH_2_	0.353 V	–	0.297 V	–	0.243 V
Ac-HVVH-NH_2_ [27]	0.389 V	–	0.339 V	–	0.235 V *
Ac-HAAH-NH_2_ [13]	0.397 V	–	0.323 V	–	0.267 V *
Ac-HADH-NH_2_	0.384 V	–	–	–	0.286 V
Ac-HDAH-NH_2_	0.384 V	–	–	0.371 V	0.288 V
	[CuHL]	[CuL]		[CuH_−1_L]	[CuH_−2_L]
Ac-HVHAH-NH_2_ [27]	–	0.253	–	–	0.181 V *
Ac-HAHPH-NH_2_ [27]	–	0.244 V	–	–	0.165 V *
Ac-HAAHGH-NH_2_ [13]	0.334 V	0.276 V	–	0.214 V	0.201 V
Ac-HAAHVH-NH_2_ [13]	0.329 V	0.264 V	–	0.202 V	0.177 V *
Ac-HAHDH-NH_2_	–	0.294 V	–	0.231 V	–
Ac-HADHAH-NH_2_	0.327 V	0.264 V	–	–	0.231 V
Ac-HDHAHDH-NH_2_	–	0.119 V	–	–	–

* Previously unpublished data.

**Table 5 molecules-27-03435-t005:** Stability constants (log *β*) and derived equilibrium constants of complexes formed in the different Ni(II)-multihistidine tetrapeptide systems (T = 298 K, I = 0.2 M KCl).

lg*β* [*M_p_L_q_H_r_*]	Ac-SarHAH-NH_2_	Ac-HAAH-NH_2_	Ac-HADH-NH_2_	Ac-HDAH-NH_2_
**[NiL]**	4.23(4)	3.90(2)	4.33(4)	4.11(3)
**[NiH_−1_L]**	–	–	−4.50(7)	–
**[NiH_−2_L]**	−13.06(5)	−13.56(3)	–	−13.71(3)
**[NiH_−3_L]**	−21.92(4)	−22.48(2)	−22.95(5)	−22.74(3)
**lg*K*(Ni(II) + 2N_Im_)**	4.23	3.90	4.33	4.11
**p*K*(amide)_1_**	–	8.73	8.83	–
**p*K*(amide)_1,2_**	8.65	–	–	8.91
**p*K*(amide)_2,3_**	–	8.92	9.27	–
**p*K*(amide)_3_**	8.86	–	–	9.03

**Table 6 molecules-27-03435-t006:** Stability constants (log *β*) and derived equilibrium constants of complexes formed in the different Ni(II)-multihistidine peptide systems (T = 298 K, I = 0.2 M KCl).

lg*β* [*M_p_L_q_H_r_*]	Ac-HFHAH-NH_2_	Ac-HAHFH-NH_2_	Ac-HKHAH-NH_2_	Ac-HAHKH-NH_2_	Ac-HDHAH-NH_2_	Ac-HAHDH-NH_2_	Ac-HAAHGH-NH_2_	Ac-HAAHVH-NH_2_	Ac-HGFHVH-NH_2_	Ac-HADHAH-NH_2_	Ac-HDHAHDH-NH_2_
**[NiH_2_L]**	–	–	20.99(3)	20.90(3)	–	–				–	18.22(6)
**[NiHL]**	10.16(4)	11.00(5)	15.20(2)	15.21(2)	11.57(8)	11.59(3)	10.70(7)	10.55(4)	11.07(3)	11.19(4)	12.05(9)
**[NiL]**	4.52(1)	5.48(3)	–	–	5.67(5)	5.55(3)	4.79(4)	4.57(3)	5.14(2)	5.15(2)	6.30(4)
**[NiL_2_]**	–	–	–	–	9.77(7)	9.37(5)	–	–	8.97(4)	–	–
**[NiH_−1_L]**	–	–	−1.79(3)	−2.10(3)	–	–	–	–	−3.19(4)	–	−2.23(7)
**[NiH_−2_L]**	−12.33(1)	−11.15(4)	−11.29(5)	−11.83(5)	−11.83(6)	−11.79(3)	−12.26(5)	−12.91(4)	–	−12.56(3)	–
**[NiH_−3_L]**	−22.09(2)	−20.71(8)	−21.45(5)	−21.76(3)	−21.30(6)	−21.89(4)	−21.21(6)	−22.62(5)	−20.21(2)	−22.29(3)	−21.51(7)
**lg*K*(Ni(II) + 2N_Im_)**	3.13	4.05	3.87	3.91	4.43	4.41	3.70	3.74	4.11	4.08	4.39
**lg*K*(Ni(II) + 3N_Im_)**	4.52	5.48	4.92	5.03	5.67	5.55	4.79	4.57	5.14	5.15	4.88
**lg*K*(Ni(II) + 4N_Im_)**	–		–	–	–	–	–	–	–	–	6.27
**p*K*(amide)_1_**	–	–	–	–	–	–	–	–	8.33	–	8.53
**p*K*(amide)_1,2_**	8.42	8.32	8.50	8.70	8.75	8.67	8.52	8.74	–	8.85	–
**p*K*(amide)_2,3_**	–	–	–	–	–	–	–	–	8.51	–	9.64
**p*K*(amide)_3_**	9.76	9.56	9.50	9.73	9.47	10.10	8.95	9.71	–	9.73	–

**Table 7 molecules-27-03435-t007:** Stability constants (log *β*) and derived equilibrium constants of complexes formed in the different Zn(II)-multihistidine peptide systems (T = 298 K, I = 0.2 M KCl).

lg*β* [*M_p_L_q_H_r_*]	Ac-HADH-NH_2_	Ac-HDAH-NH_2_	Ac-HAHDH-NH_2_	Ac-HDHAH-NH_2_	Ac-HADHAH-NH_2_
**[ZnHL]**	–	–	–	10.84(8)	10.91(5)
**[ZnL]**	4.41(7)	4.01(3)	5.14(5)	5.38(3)	5.63(1)
**[ZnH_−1_L]**	−3.13(12)	−3.49(5)	–	–	−2.40(6)
**[ZnH_−2_L]**	−11.69(12)	–	–	–	−11.23(6)
**lg*K*(Zn(II) + 2N_Im_)**	4.41	4.01	–	3.70	3.80
**lg*K*(Zn(II) + 3N_Im_)**	–	–	5.14	5.38	5.63
**lg*K*(ZnL/ZnH_−1_L)**	7.54	7.50	–	–	8.03
**lg*K*(ZnH_−1_L/ZnH_−2_L)**	8.56	–	–	–	8.83

## Data Availability

Data is contained within the article.

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
