# Peer review of "The Role of Side Chains in the Fine-Tuning of the Metal-Binding Ability of Multihistidine Peptides"

_molecules, 2022, doi:10.3390/molecules27113435_

Round 1
Reviewer 1 Report
In this paper, the author systematically studied the copper(II), nickel(II) and zinc(II) complexes of multihistidine peptides containing amino acids with different side chains, the results showed the side chains of different amino acids affect the complex formation processes and the electrochemical parameters. This is meaningful and interesting work. I think it can be published with a few minor revisions.
The article is a bit verbose, so it is recommended to put some diagrams in the supporting information, such as Table1. Some of the tables are unreadable, such as Table2 and Table3.
The caption of the Figure should be carefully checked. The “b” was missing in the captions of Figure 2 and 4, and the “Figure 2c” in page 9 is wrong.
Author Response
Many thanks the Referee for the careful evaluation of the manuscript and the valuable comments. We have corrected and completed the text and figures of manuscript and transferred the Table 1 and Table 9 to the Appendix. I hope the tables on the landscape page will be visible in the paper after editing in the journal.
Reviewer 2 Report
The paper describes a systematic study on the complex-formation equilibria between three divalent metal ion and several poly-histidine peptides, aimed at clarifying the role played in the chelation process by the number and the position of both His residues and other amino acids, as Phe and Asp. The investigation is performed by using and comparing many experimental techniques and it is supported by a relevant bibliographic section. The results are well described, and the conclusions are sound. Therefore, I suggest accepting this paper after only a minor revision.
In particular:
-page 2, line 31: can behave (instead of can behaves)
-page 2: at the end of the introduction a short sentence with an overview of the work and the the aim of the study should be inserted
-Table 1 and following lines: Ac-S3H4-NH2 should be written Ac-SSSHHHH-NH2, for homogeneity with the other peptides and for the sake of clarity
-page 5, last line: logK data ere reported in Table 3 (not Table 2); in Fig. 1 logB([CuL]) are shown, not logK
-Fig. 1 - The square brackets are not in the right position
-Fig. 1 – The data referring to the 4xNim complex of Ac-SarHaH-NH2 shouldn’t be reported here, because it concerns a 1:2 complex and not a [CuL] species, as reported in the ordinate axis
-Fig. 2, caption: Cu(Ac-HDHAH-NH2 is misprinted; (b) should be specified
-three lines after Fig.2: a comma should be written after [CuL] (instead of a period)
-page 9, line 3: Figure 3b (not 2b)
-page 9 line 17: Figure 3c (not 2c)
-page 12, last line: the sentence “their cyclic voltammetric measurements was also performed” should be rewritten
-page 13, line 4: “submitted” should be substituted by reported or shown
-page 13, line 7: eps° shouldn’t be E°?
-Table 5, caption: the Reference(s) for the “previously published data” should be specified
-page 14, ten lines to the end: “This suggests that copper(II) binding to the –HXZH– sequence….” shouldn’t be “nickel(II)”?
-Par. 3.3 Ni(II) complexes: what about the kinetics? Normally the coordination of amide nitrogens is accompanied by the transition from octahedral and high spin to square planar and low spin geometry (yellow species) and this process is slow. How was the complex-formation kinetics of Ni(II) taken into account?
-page 19, beginning: it is not clear if the absence of precipitation was observed for the -HADH- sequence only (as written in the first line) or also for the sequence -HDAH-, as it could be inferred from Fig. 13
-page 19, line 7: Ref. 26 should be 27.
Author Response
Many thanks the Referee for the careful evaluation of the manuscript and the valuable comments. We have corrected and completed the text and figures of manuscript as it was suggested.
The answers to some of the comments and questions are as follows:
-page 2: at the end of the introduction a short sentence with an overview of the work and the the aim of the study should be inserted
The introduction is completed.
-Table 1 and following lines: Ac-S3H4-NH2 should be written Ac-SSSHHHH-NH2, for homogeneity with the other peptides and for the sake of clarity
The mentioned peptide is Ac-His-Sar-His-Sar-His-Sar-His-NH2, the abbrevation was used in the ref. 8. An explanatory note was added to Table 1, Table A1 and A2.
-page 5, last line: logK data ere reported in Table 3 (not Table 2); in Fig. 1 logB([CuL]) are shown, not logK
-Fig. 1 – The data referring to the 4xNim complex of Ac-SarHaH-NH2 shouldn’t be reported here, because it concerns a 1:2 complex and not a [CuL] species, as reported in the ordinate axis
Referee is right, the X-axis title and caption in Figure 1 are not correct. The calculated log K(Cu(II) + z (Nim)) values are plotted (not log beta (CuL) values)). Therefore Ac-SarHAH-NH2 is seen at 4xNim coordinated complexes, log beta(CuL2) corresponds to log K(Cu(II) + 4Nim).
-Table 5, caption: the Reference(s) for the “previously published data” should be specified
Redox potential values ​​of the CuL and CuL2 complexes of the tetrapeptides have been reported in the Ref. 11, but the redox potential values ​​of CuLH-2 were measured only after publication. That is why the comment is in the title of the Table 4 (Table 5): "previously unpublished data".
-Par. 3.3 Ni(II) complexes: what about the kinetics? Normally the coordination of amide nitrogens is accompanied by the transition from octahedral and high spin to square planar and low spin geometry (yellow species) and this process is slow. How was the complex-formation kinetics of Ni(II) taken into account?
Referee is right. The formation of amide nitrogen coordinated nickel(II) complexes is slow, but the rate of complex formation depends on the type and length of the peptide. In general, in the case of N-terminally free penta- or longer peptides, the formation of square planar complexes from octahedral ones is very slow and practically impossible to study by potentiometry. However, the complex formation processes of tetra- or shorter peptides can be characterized using potentiometry (a lot of data thermodinamic stability constants are available in the literature). For terminally protected peptides, where the histidine(s) is/are the anchor group(s), the square planar nickel(II) complexes are formed more rapidly, so that the equilibrium of complex formation can be monitored by potentiometry. For this reason, the kinetics of complex formation were not taken into account.
Reviewer 3 Report
The manuscript by Katalin Várnagy and co-authors is a study on the influence of amino acid side chains in multihistidine peptides on binding with Cu, Ni and Zn ions. In the study, different techniques are applied to a large number of peptide complexes to obtain a clear picture of the coordination modes. The conclusions are consistent with the observations, so overall this is a good paper for publication in Molecules.
Small error: authors underlined in ref 4
Author Response
Many thanks the Referee for the careful evaluation of the manuscript. The text is corrected.
Reviewer 4 Report
This manuscript submitted by Várnagy et al. reports about the metal-binding properties of peptides containing two or more His amino acids. The paper is a systematic and thorough study about metallopeptides, and this reviewer thus recommends acceptance of the paper with only minor revisions after having considered the following two points.
- There are figures/tables with numbers that refer to an appendix, but this referee did not have access to them.
- There are too many self-citations in comparison to references referring to work of other groups (8 out of 28). This ratio should be improved.
Minor points:
p. 1: The studied metal complexes should appear in the abstract.
p. 2: The use of references 7 and 11 is not clear. They appear after the very first sentence of the paragraph, however, they relate (probably) to the whole paragraph, and should thus appear after the last sentence of the respective paragraph.
p. 4 silica cells --> Should better be called quartz cells.
p. 9, Figure 3: As for the other examples, the sequence should be provided also for c).
p. 11, first paragraph: How was the existence of dinuclear complexes proven?
p. 12, last paragraph: It is assumed that aspartic acid might have an effect on the redox potentials of the Cu(II) complexes. A hypothesis of why this should be the case should be provided.
p. 13, bottom: “formal potential values” --> It is not clear what this expression means.
p. 14, top: The electrochemical parameters are rationalized with a distortion of the coordination geometry --> An explanation and a reference should be provided.
p. 15, Table 6 (and also at other places of the manuscript) --> “amid” should read “amide”
p. 22, Reference 21: The cited link does not work. Please provide the correct link.
Author Response
Many thanks the Referee for the careful evaluation of the manuscript and the valuable comments. We have corrected and completed the text and figures of manuscript as it was suggested.
- There are too many self-citations in comparison to references referring to work of other groups (8 out of 28). This ratio should be improved.
The studies of metal complexes of multihistidine peptides has been going on in our group for 15 years, and numerous papers were published in this research field. Since one of the aims of the present work is to collect equilibrium, spectroscopic and electrochemical data of multihistidine peptides containing different side chain donor groups and to compare them with the previously studied multihistidine peptides, we used several self-citations.
Minor points:
- 1: The studied metal complexes should appear in the abstract.
- 2: The use of references 7 and 11 is not clear. They appear after the very first sentence of the paragraph, however, they relate (probably) to the whole paragraph, and should thus appear after the last sentence of the respective paragraph.
- 4 silica cells --> Should better be called quartz cells.
- 9, Figure 3: As for the other examples, the sequence should be provided also for c).
The text and captions in the figures have been modified in accordance with the comments above.
- 11, first paragraph: How was the existence of dinuclear complexes proven?
On the one hand, no precipitation was observed in the samples containing excess of metal ion, even in alkaline medium. On the other hand, the CD spectra of different copper(II)-ligand ratios support the formation of polinuclear species. See below the Figure A3 in Appendix. (See attached file.)
- 12, last paragraph: It is assumed that aspartic acid might have an effect on the redox potentials of the Cu(II) complexes. A hypothesis of why this should be the case should be provided.
- 14, top: The electrochemical parameters are rationalized with a distortion of the coordination geometry --> An explanation and a reference should be provided.
We have improved the explanation:
Studies of Cu(II) complexes with different type of ligands have shown that the rigid structure of the Cu(II) complex cannot satisfy the coordination requirement of Cu(I). This explains that copper(II) complexes with quasi square planar geometry (NH2,N–,N–,N–) coordinated species of oligopeptides, (NH2,N–,N–,NIm) coordinated complex of GGH peptide, (N–,N–,N–,NIm) coordinated complexes of terminally protected histidine containing peptides) has very low (negative) redoxipotential values. The reduction process of Cu(II) complexes has been proved to be facilitated when the Cu species can easily evolve toward a tetrahedral geometry upon reduction, which increases the stability of Cu(I) complex.[28-33]
Based on these facts, the redox potential values fall in the positive range due to the distorted geometry of the [CuH-2L] complex of multihistidine peptides and the geometry of the complex is further distorted by the interaction between metal center and the carboxylate group of aspartic acid.
- 13, bottom: “formal potential values” --> It is not clear what this expression means.
- 15, Table 6 (and also at other places of the manuscript) --> “amid” should read “amide”
- 22, Reference 21: The cited link does not work. Please provide the correct link.
The text and reference 21 were corrected.
